

# Diverse microbial communities hosted by the model carnivorous pitcher plant *Sarracenia purpurea*: analysis of both bacterial and eukaryotic composition across distinct host plant populations

Jacob J. Grothjan[1] and Erica B. Young[1,2]

[1] Department of Biological Sciences, University of Wisconsin-Milwaukee, Milwaukee, WI, United States of America
[2] School of Freshwater Sciences, University of Wisconsin-Milwaukee, Milwaukee, WI, United States of America

Corresponding author
Erica B. Young, ebyoung@uwm.edu

## ABSTRACT

**Background**. The pitcher plant *Sarracenia purpurea* supplements nutrient acquisition through carnivory, capturing insect prey which are digested by a food web community of eukaryotes and bacteria. While the food web invertebrates are well studied, and some recent studies have characterized bacteria, detailed genetic analysis of eukaryotic diversity is lacking. This study aimed to compare eukaryotic and bacterial composition and diversity of pitcher communities within and between populations of host plants in nearby but distinct wetland habitats, and to characterize microbial functions across populations and in comparison with another freshwater community.

**Methods**. Pitcher fluid was sampled from the two wetlands, Cedarburg and Sapa Bogs, community DNA was extracted, and 16S and 18S rRNA amplicons were sequenced and data processed for community-level comparisons.

**Results and Conclusions**. Bacterial diversity in the small pitcher volume rivaled that of larger aquatic communities. Between pitcher plant populations, several bacterial families (Kiloniellaceae, Acetobacteraceae, Xanthobacteraceae, Sanguibacteraceae, Oligoflexaceae, Nitrosomonadaceae, Chromatiaceae, Saprospiraceae) were significantly higher in one population. However, although predicted pitcher bacterial functions were distinct from other freshwater communities, especially for some amino acid metabolism, functions were similar across all the pitchers in the two populations. This suggests some functional redundancy among bacterial taxa, and that functions converge to achieve similar food web processes. The sequencing identified a previously under-appreciated high diversity of ciliates, Acari mites, fungi and flagellates in pitcher communities; the most abundant sequences from eukaryotic taxa were Oligohymenophorea ciliates, millipedes and Ichthyosporea flagellates. Two thirds of taxa were identified as food web inhabitants and less than one third as prey organisms. Although eukaryotic composition was not significantly different between populations, there were different species of core taxonomic groups present in different pitchers—these differences may be driven by wetland habitats providing different populations to colonize new pitchers. Eukaryotic composition was more variable than bacterial composition, and there was a poor relationship between bacterial and eukaryotic composition within individual pitchers, suggesting that colonization by eukaryotes may be more stochastic than for

bacteria, and bacterial recruitment to pitchers may involve factors other than prey capture and colonization by eukaryotic food web inhabitants.

## INTRODUCTION

Carnivorous plants grow in nutrient deficient wetland environments and use carnivory to supplement their mineral nutrition, and include Venus fly traps, sundews, and pitcher plants (*Ellison & Adamec, 2018*). The northern or purple pitcher plant, *Sarracenia purpurea* subsp. *purpurea* L. produces modified leaves formed into pitchers that function as passive pitfall traps which fill with rainwater and capture insect prey. Breakdown of insect prey in *S. purpurea* is achieved by an inquiline (living in the pitcher habitat) food web of invertebrates and microbes which colonize after pitcher opening (*Peterson et al., 2008*). The microbes produce digestive enzymes to release nutrients from organic prey particles to support the food web and provide nutrients for host plant uptake (*Young, Sielicki & Grothjan, 2018*). Core invertebrate trophic levels of the food web are well characterized (*Gotelli & Ellison, 2006*; *Mouquet et al., 2008*) and recent studies provide some insights into the bacteria present (*Gray et al., 2012*) but we have limited understanding of the functions of microbial communities or diversity of eukaryotic microbes represented within the *S. purpurea* pitcher plant food web.

When pitchers first open, they are sterile and need to be colonized by bacteria and eukaryotes (*Peterson et al., 2008*), then the pitcher community changes over time with succession, and may be regulated by resource availability (*Miller & TerHorst, 2012*; *Gray et al., 2012*; *Armitage, 2017*). The invertebrate taxa of the food web have several known functions; Sarcophagid fly larvae shred insect prey and stir the fluid, midges live at the base of the pitcher, breaking apart detritus, and rotifers and ciliates are mid-trophic level grazers (*Bledzki & Ellison, 1998*; *Paisie, Miller & Mason, 2014*). The mosquito larvae, *Wyeomyia smithii* is a top predator grazer in pitcher plant food webs (*Kneitel & Miller, 2002*; *Baiser et al., 2013*). Autotrophic algae may be present in pitchers (*Gebühr et al., 2006*). The composition of pitcher inhabitants is known to vary between individual pitchers and across plant populations (*Peterson et al., 2008*; *Gray, 2012*), but little is known about the relationship between eukaryotic and bacterial composition of the communities, between pitchers and across pitcher plant populations. While genetic sequencing has provided detailed information about bacterial communities in *Sarracenia* communities (*Koopman & Carstens, 2011*; *Gray et al., 2012*) and limited genetic analysis has targeted eukaryotes in other carnivorous plant species (*TerHorst, 2011*; *Bittleston et al., 2016*; *Satler, Zellmer & Carstens, 2016*), detailed genetic analysis of the eukaryotic communities using rRNA targets has not previously been applied to *S. purpurea* pitcher plant communities.

Microbial composition may vary with presence of other food web members and with geographical distance (*Koopman & Carstens, 2011*; *Paisie, Miller & Mason, 2014*; *Bittleston*

*et al., 2018*), and can be related to host habitat (*Krieger & Kourtev, 2012*; *Satler, Zellmer & Carstens, 2016*), but how much microbial functions vary with host plant habitat or population is less clear. Pitcher plants show acclimation to wetland habitats, for example, as changes in morphology related to carnivory in response to nutrient availability (*Gotelli & Ellison, 2002*). Two nearby wetlands in Wisconsin offer different nutrient availability, pH and surrounding vegetation conditions which can induce morphological changes in *S. purpurea*. In Sapa Bog, lower pH, higher nitrogen and phosphorus in the soil water resulted in narrower pitchers than Cedarburg Bog pitcher plants (*Bott, Meyer & Young, 2008*). These two distinct habitats and populations are ideal to compare food web composition, microbial diversity and function in different *S. purpurea* populations in response to distinct habitat conditions.

Early isolation and culturing of pitcher plant bacteria identified a range of microbial functions (*Lindquist, 1975*), and other studies applied genetic analysis techniques to cultured isolates (*Whitman et al., 2005*; *Siragusa, Swenson & Casamatta, 2007*) or identified particular functional groups of pitcher plant bacteria (*Young, Sielicki & Grothjan, 2018*). Recent studies have linked enzyme activity to bacterial functions in nutrient cycling within *S. purpurea* pitchers (*Luciano & Newell, 2017*; *Bittleston et al., 2018*; *Young, Sielicki & Grothjan, 2018*). However, despite studies of bacterial diversity, characterization of bacterial enzyme functions, the critical importance of bacteria in prey digestion and thus carbon and nutrient supply to the food web, bacteria have only recently been considered as more than a 'black box' in food web models (*Lau et al., 2018*). It is unknown if the microbial functions of the pitcher plant detrital food web are similar to or distinct from other aquatic ecosystems. To understand what functions are specific to these food webs and to expand the inclusion of bacterial functions in this model food web system, more detailed information about pitcher plant bacterial community metabolic functions is needed.

To address these gaps in our understanding of eukaryotic diversity and bacterial functions in pitcher plant communities and the variability between pitcher microbial populations, this study applied mass gene sequencing and metagenomic functional predictions to compare microbial communities within two distinct populations of *S. purpurea*. This study aimed to address the following specific research questions:

(1) How does the bacterial and eukaryotic taxa composition within *Sarracenia purpurea* pitchers differ between pitchers within the same population and between two populations in nearby but distinct wetland habitats?

(2) How does the bacterial and eukaryotic diversity vary between *S. purpurea* pitchers in the two plant populations?

(3) How do the predicted functions of the bacterial community compare between the plant populations and with another freshwater community?

## MATERIALS AND METHODS

### Site descriptions, plant selection, and sampling

Two populations of the pitcher plant *Sarracenia purpurea* subsp. *purpurea* (hereafter *S. purpurea*) were sampled in June, 2013, from two distinct wetlands, ~1.5 km apart,

separated by farmland. One population was in the Cedarburg Bog (43°23.2′N, 88°0.63′W), a peatland fen in SE Wisconsin characterized by low lying marshy areas interspersed with elevated patches of cedar (*Thuja occidentalis*) and tamarack (*Larix laricina*). *S. purpurea* grows in slightly elevated areas with sphagnum moss or open marshy areas and the study site was accessed by a boardwalk. The second population was in an ombrotrophic bog, Sapa Bog (43°23.64′N, 88°1.4′W) characterized by a dense growth of black spruce (*Picea mariana*) and tamarack, providing a shadier canopy than in the Cedarburg Bog (*Bott, Meyer & Young, 2008*). The two populations experience different growth conditions with more acidic soil substratum and higher plant available nitrogen (N) and phosphorus (P) in Sapa Bog (*Bott, Meyer & Young, 2008*). A field permit was issued by the University of Wisconsin-Milwaukee Field Station committee.

*S. purpurea* pitchers were selected using the following criteria: (1) entirely green, avoiding any sign of damage or senescence (2) with aperture >3 cm to allow sampling with a syringe and (3) the fluid contained suspended particles indicating active detrital and food web processes. Five samples were collected from each of the two wetland population for a total of 10 samples. Some samples were from single pitchers, others were combined samples from up to 3 separate pitchers to reach sample volume of ≥30 mL. Pitcher fluid was collected with a sterilized syringe and tubing inserted into the pitcher, and the fluid mixed prior to sampling by drawing fluid into the syringe and dispelling (*Young, Sielicki & Grothjan, 2018*). Samples were transferred to sterile 50 mL tubes, stored on ice, and transported to the lab. Samples were pre-filtered through 153 μm mesh (Sefar NITEX, Montreal) to remove large debris, and vacuum filtered onto 0.2 μm polycarbonate membrane filters (GE Water and Process Technologies, Pennsylvania) which were stored at −70 °C.

## DNA extraction and sequencing

Frozen filters with cells and particles were used for extraction of total community DNA using a FAST DNA soil extraction kit (MP Biomedicals, Santa Ana, CA, USA). DNA concentration and purity were confirmed with agarose gel electrophoresis and spectrophotometry (NanoDrop ND-1000). To examine the bacterial and eukaryotic organisms represented in the pitcher fluid using genetic analysis, a range of PCR primers were tested for amplification of sequences from the total community DNA, targeting the 16S and 18S rRNA genes. Gene targets producing the clearest and most consistent bands in agarose gel electrophoresis were selected for use in DNA sequencing—16S F338 (5′-ACTCCTACGGRAGGCAGCAG-3′) (*Dethlefsen et al., 2008*) and R802 (5′-TACNVGGGTATCTAATCC-3′) (*Claesson et al., 2010*) and 18S F426 (5′-TCCAAGGAAGGCAGCAGG-3′) and R853 (5′-AGTCCTATTCCATTATTCCATG-3′) (*Marron, Akam & Walker, 2013*). Samples were sequenced by the Great Lakes Genomic Center at University of Wisconsin-Milwaukee using 2 × 250 bp sequencing runs on an Illumina MiSeq using the 16S and 18S rRNA primers with Illumina adapters and manufacturer protocols. The sequence depth was for 16S 43–110 K reads and for 18S was 99–147 K reads.

## Sequence analysis and bioinformatics

16S rRNA V3_4 region sequences were analyzed with mothur version 1.33.3 with MiSeq SOP (accessed August 2014; *Kozich et al., 2013*) and with QIIME (MacQIIME V1.9.1). Quality control was performed by eliminating sequences with low quality scores and filtering chimeric sequences from samples using usearch (version 5.2.236; *Edgar, 2010*). Alignment and analysis of 16S sequences for taxonomic identity used the SILVA SSU database (Release 119, 123 (*Quast et al., 2013*)). Bacterial taxon richness and diversity were quantified by clustering sequences with a 97% similarity OTU definition (*Chao, 1984*) using a *de novo* approach via the uclust algorithm (version v6.1.544) where individual sequences are treated as "seeds" from which to build clusters (*Edgar, 2010*).

18S rRNA V3_V4 sequences were analyzed in QIIME with SILVA SSU (release 123). Unzipped Forward (F) and Reverse (R) fastq data output files were joined (quality parameters $j = 75$ and $p = 0.9$) using fastq-join (*Aronesty, 2011*) in QIIME. Joined sequences that did not meet the minimum length threshold (464 bp for 16S and 427 bp for 18S) were removed. Sequences were separated from their quality scores and formatted for downstream processing using sed within UNIX command line. Chimeric sequences were removed via usearch v6.1 (*Edgar, 2010*) utilizing comparison with SILVA reference database (Release 123) for chimera detection and removal (*Haas et al., 2011*). Reads were clustered using a 97% similarity OTU definition (*Chao, 1984*) using a *de novo* approach via the uclust algorithm (*Edgar et al., 2011*) and sequences aligned using PyNAST (*Caporaso et al., 2010*), assigned taxonomy with the BLAST algorithm (*Edgar, 2010*), using SILVA.

## Community comparisons

For comparison of the two plant populations, rarefaction curves were generated in QIIME and the QIIME ANOSIM command was used to statistically test for similarities in community composition between populations. Additional QIIME commands aided in downstream sequence analysis including eliminating singletons from samples and using taxonomic composition to generate alpha and beta diversity indices, and unifrac values for Principle Coordinate Analysis (PCoA) (*Lozupone et al., 2011*) with plotting and visualization in PAST (*Hammer, Harper & Ryan, 2001*) or for building sample relatedness trees using FastTree 2.1 in QIIME (*Price, Dehal & Arkin, 2010*). The taxa present in significantly different frequencies in the two populations were identified using 1-way ANOVA of the 5 samples from each population using Sigmaplot (v12.5, Systat Software Inc, San Jose, CA, USA). Diversity metrics were also compared between populations using 1-way ANOVA in Sigmaplot. Pitcher community composition was compared in PCoA and relatedness trees using outgroups generated from freshwater wastewater bacterial communities (*Xiao et al., 2015*) for 16S, and freshwater eukaryotic database (EUKBASE) created from SILVA NR108 (*Pruesse et al., 2007*) for 18S rRNA sequences. 16S rRNA sequences and taxon identities were used to predict metagenomic functional gene categories using PICRUSt based on KEGG biochemical pathways (*Langille et al., 2013*) with weighted NSTI scores of 0.03–0.39 for pitchers and 0.12–0.33 for the comparison freshwater community. PICRUSt functional data was used for PCoA analysis using HUMAnN2 v0.11.1

(*Abubucker et al., 2012*), and functional gene category predictions were compared across samples and with the wastewater bacterial communities.

## RESULTS

### Bacterial and eukaryotic taxa in pitcher fluid

The bacterial and eukaryotic community composition of individual pitchers varied, with some pitchers very similar in composition, others distinct (Fig. 1). Based on sequence identity, the bacterial families were typically represented more evenly within each pitcher than Eukaryotic families, as most pitcher samples were dominated by relatively few Eukaryotic families, though the dominant Eukaryotic families varied across pitchers (Fig. 1). Pitchers sampled in Cedarburg were more similar in bacterial composition than Sapa pitchers, and CB1p1 and CB1p2 are nearly identical in both bacterial and eukaryotic composition (Fig. 1). Some Sapa pitcher communities showed dominance of a single taxon, for example Sp3p4 was dominated by the bacterial family Coxiellaceae, and eukaryotic sequences were dominated by the ciliate taxon Scuticociliatia of which 84.6% was contributed from a single OTU. In other samples, single dominant taxa were comprised of several OTUs; in CB1p1 96.8% of the sequences identified as Pseudomonadaceae were contributed from 5 OTUs. Taxonomic composition pooled for each wetland (Fig. S1), showed distinctions between the two populations, with more even representation of bacterial families in Cedarburg than in Sapa pitchers in which >50% of sequences were attributed to 4 families (Fig. S1). In contrast, the pooled composition of Eukaryotic families for Cedarburg was dominated by 4 families, while Sapa communities showed more even representation of Eukaryotic families.

Comparison of composition of the most abundant bacterial and eukaryotic taxa between wetlands (Fig. 2), showed many common bacterial and eukaryotic families but with few families present in every sample (i.e., at least one sample showed 0 abundance). The heatmaps also illustrate that Cedarburg samples showed a greater number of common bacterial and eukaryotic OTUs between samples than Sapa samples (Fig. 2, Fig. S2). Bacterial composition of pitchers was dominated by groups Saccharibacteria (formerly candidate division TM7), $\alpha$- $\beta$- and $\gamma$-Proteobacteria, Bacteroidetes, Flavobacteria, and Firmicutes (Fig. 2). There were 10 bacterial families which were more abundant in one wetland population (ANOVA, $p < 0.05$, Fig. 2). Of the 15 most common bacterial families, only Neisseriaceae was significantly more abundant in Sapa samples ($p < 0.02$) while 9 other less common bacterial families differed significantly between the two populations—Kiloniellaceae was also higher in Sapa samples ($p < 0.035$) and Acetobacteraceae, Rhizobiales AT, Xanthobacteraceae, Sanguibacteraceae, Oligoflexaceae, Nitrosomonadaceae, Chromatiaceae, Saprospiraceae were all higher in Cedarburg samples ($p < 0.05$). Across all pitchers, the highest abundance bacterial OTUs identified to genus included *Rickettsiella, Azospirillum, Pedobacter, Pseudomonas, Aquitalea, Sphingomonas, Duganella,* and *Alkanidiges* (Table S1).

Eukaryotic taxa present in the pitchers included different families of ciliates, millipedes, springtails, midges, insects, fungi, flagellates, and other protists (Fig. 2, Table S2). Despite

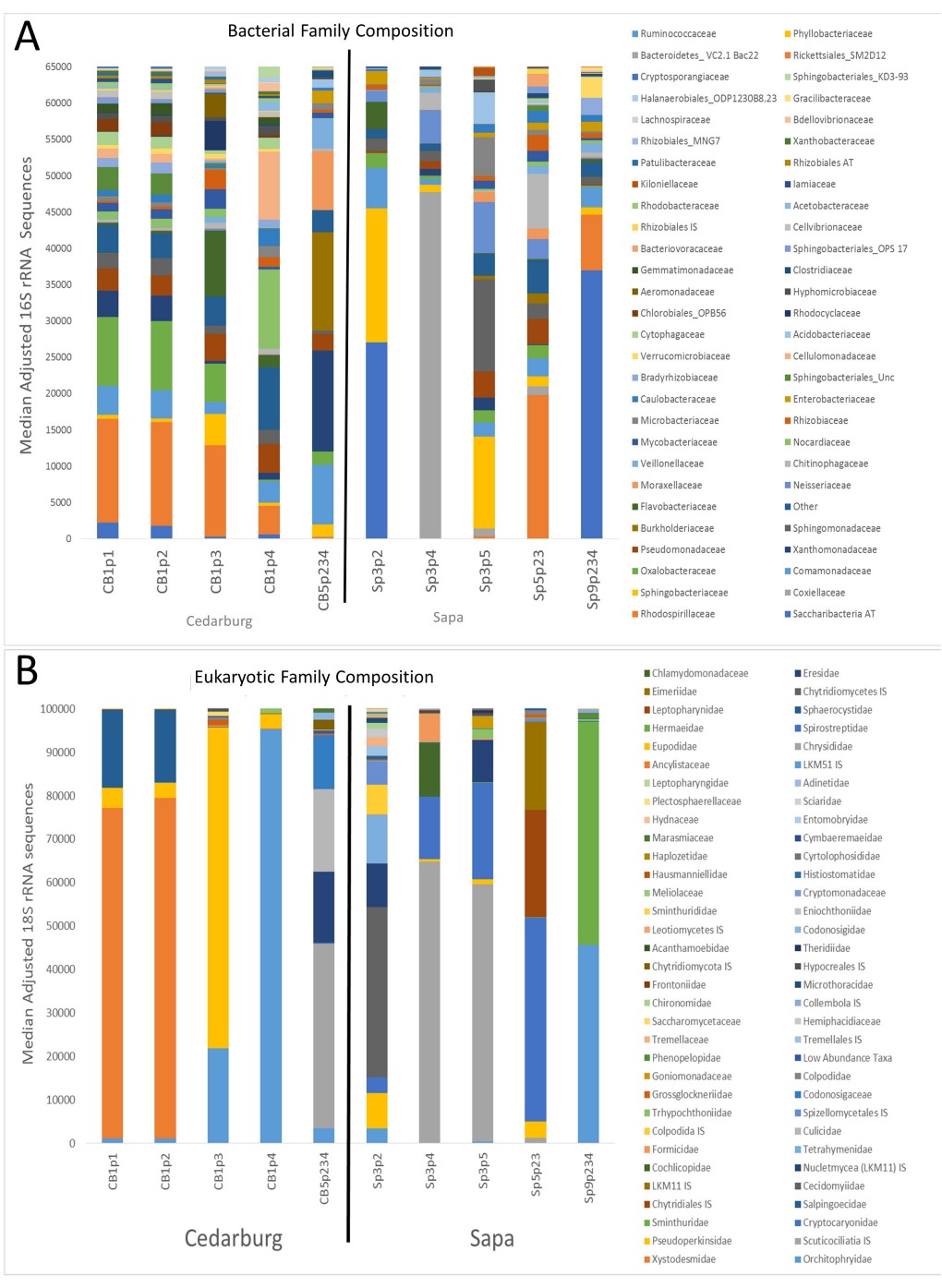

**Figure 1** **Bacterial and eukaryotic pitcher plant composition of samples in two populations.** Genetic analysis of pitcher plant communities from two wetland populations, Cedarburg and Sapa. Family phylotypes are based on 16S rRNA (bacterial) (A) and 18S rRNA (Eukaryotic) (B) taxa identified from sequencing of total community DNA in pitcher fluid samples. Samples were collected from Cedarburg (CB) and Sapa (Sp) wetlands from single plants (e.g., CB1-5) from individual pitchers (e.g., p1-4) or combined pitchers (e.g., p234). Taxa with only one representative sequence were removed, and any taxa representing < 0.01% of total in each sample were pooled as 'Other'. AT, ambiguous taxa, IS, incertae sedis, unc., uncultured.

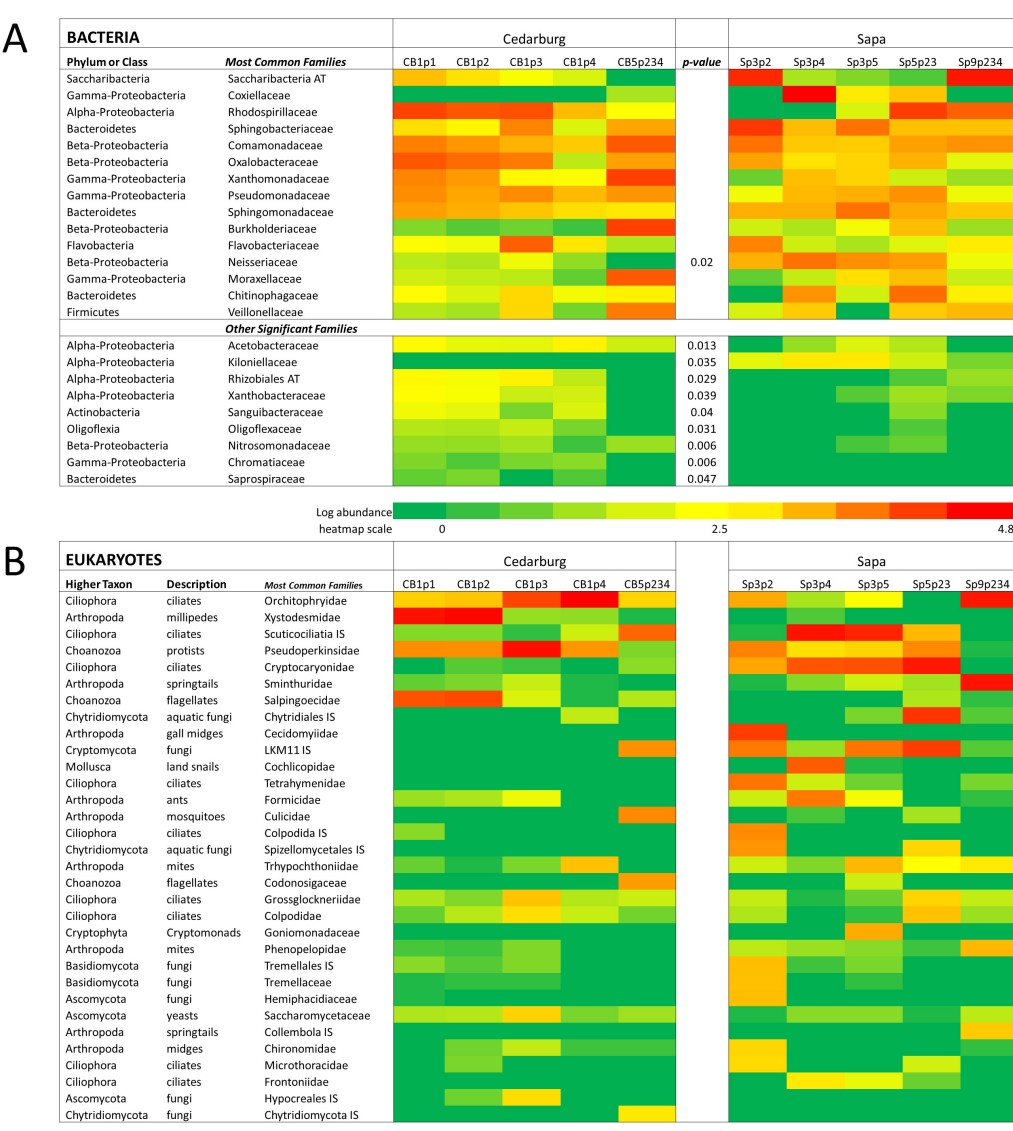

**Figure 2  Heatmap of bacterial and eukaryotic families across the samples in two populations.** Heatmap of family-level taxa based on 16S rRNA (A) and 18S rRNA (B) sequences isolated from pitcher fluid samples (not-median adjusted). Color scale corresponds to the logarithmic transformation of the number of times a taxon was observed in each sample (green is lowest abundance, red is highest abundance). For bacteria in five samples from each population, taxa which were significantly higher in one population are shown with a *p*-value. Abbreviations are as for Fig. 1.

some compositional differences in Eukaryotes between wetlands (Fig. S1), none of the Eukaryotic families were found to be significantly more abundant in either wetland population (ANOVA, $p > 0.05$, Fig. 2). However, clearly some taxa were dominant in some samples, but absent from others, and there were many more Eukaryotic families only represented in one or two samples (e.g., Chytrid families, Gall midges—Cecidomyiidae, land snails—Cochlicopidae, the Cryptomonad—Goniomonadaceae, and Collembola— springtails). Detailed data on genus- and species-level identifications of eukaryotes present

in each sample are included in the supplementary materials (Table S2). Of the 25 most abundant taxa identified, eight were ciliate taxa, 8 were Arthropods (three Acari mites, three insects), five were fungi, and three were Opisthokont flagellates (Table S2). The two most common taxa identified were similar to an Orchitophyridae ciliate environmental taxa (found in all 10 samples), and a Scuticociliata uncultured taxon (found in nine of 10 samples). The millipede species *Cherokia georgiana* was the most abundant genus, present in five of 10 samples, an opisthokont flagellate Ichthyophonida_LKM51 was identified in all 10 samples. Many taxa were only present in samples from one wetland but when major groups were considered (Fig. 3), there were more arthropod and flagellate sequences in Cedarburg samples but more ciliates in Sapa, although across the 5 samples in each wetland, there were no statistical differences between abundance in Cedarburg vs Sapa (1-way ANOVA). Two different mosquito taxa of the family Culicidae were identified in the two wetland populations (Table S2). The freshwater bdelloid rotifer genus *Adineta* was represented in the similar Cedarburg CB1p1 and CB1p2 samples (Fig. 1), but not in any other samples. Gastropod sequences were only found in 2 Sapa pitchers from the same plant, and tardigrade DNA was found in one Sapa sample (Table S2). Algae and plant sequences were present in very low abundance in Sapa and were absent from Cedarburg pitchers. When Eukaryotic families were scored as probable food web members vs prey, vs other, based on literature on habitat and organism functions (Fig. 3), 67% of sequences were identified as food web, 28% of sequences were probable prey and 5% were ambiguous, potentially present incidentally (e.g., fungal spores or plant parts possibly fallen into pitchers, taxa with unknown habitat or ecological role, or taxa of too broad a classification to make grouping the organisms possible).

## Diversity analysis

Rarefaction curves for these samples are shown in Fig. S2, with no consistent coverage for samples between 16S and 18S sequencing. Good's coverage estimate was high (>0.95) across most samples, except CB1p4 (Table 1). In bacterial diversity, Cedarburg samples showed significantly higher number of OTUs, Chao1 richness, and ACE diversity, but also more singletons, than Sapa samples (1-way ANOVA, $p < 0.025$; Table 1). Except for Sp5p23, all Cedarburg samples had more bacterial OTUs than Sapa samples. The least diverse samples in terms of bacteria were from Sapa (Sp3p2, Sp3p4, Sp9p234) but for eukaryotic composition, Cedarburg had the least diverse (CB1p1, CB1p2, CB1p3 and CB1p4) as well as the most diverse sample (CB5p234). There were no significant differences in number of OTUs or diversity metrics between the two populations of eukaryotes (Table 1).

## Community comparisons and functional predictions

PCoAs were based on bacterial composition using unifrac weighted jackknife settings, and metagenome predictions from 16S rRNA-based taxonomic composition through PICRUSt (Fig. 4A). There was close overlap in bacterial composition between Cedarburg and Sapa samples with no distinct differences (ANOSIM $p > 0.05$), but the wastewater community used as an outgroup was distinct. Sp3p2, Sp3p4, and Sp9p234 were more distant from the tight clustering of other pitcher samples.

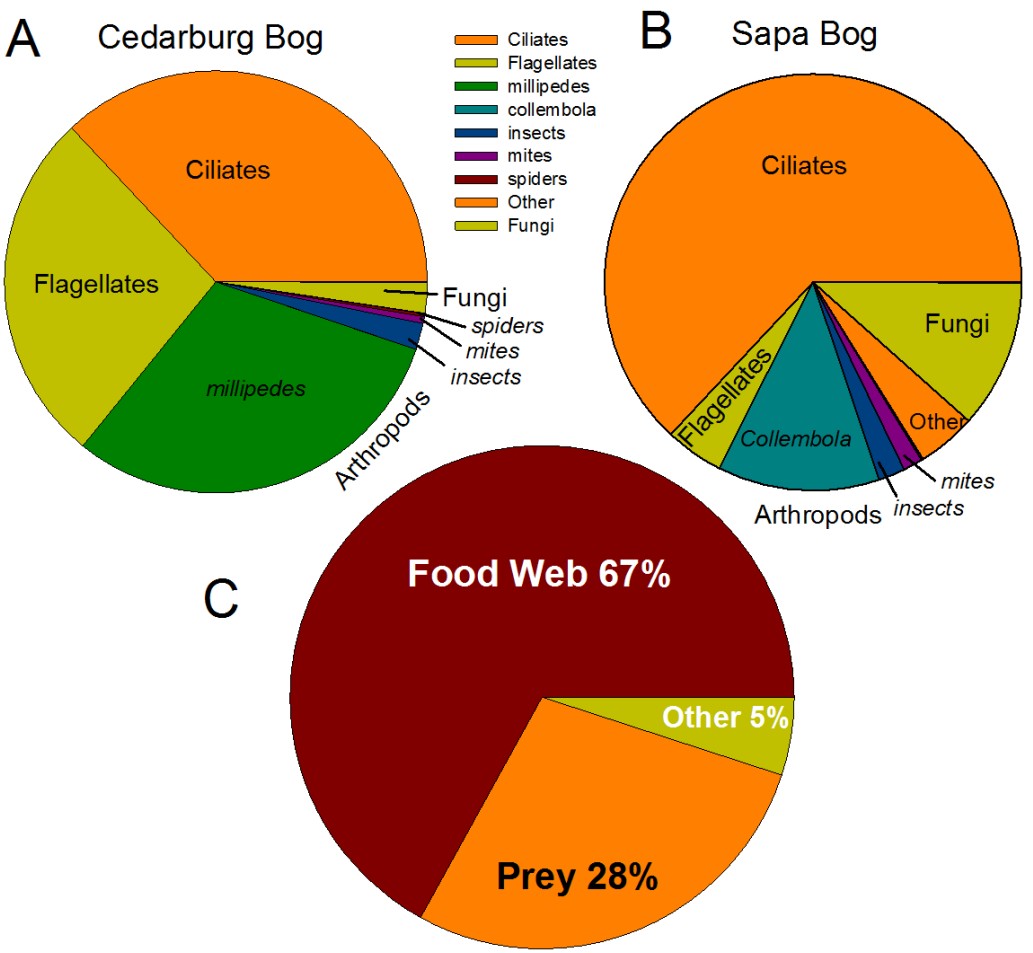

**Figure 3** **Pie charts of major eukaryotic groups and roles in microecosystem.** Composition of eukaryotic groups based on 18S rRNA sequences, showing most common Arthropod groups, ciliates, (ospithokont) flagellates, fungi, and other groups for each wetland population (A, B) (full details of taxa in each sample provided in Table S2). (C) Using data pooled for all 10 samples, taxa representing >1% of total sequences were scored as **Food Web** for aquatic taxa or those known to be pitcher inhabitants, **Prey** for non-aquatic Arthropods as likely captured prey, or **Other** for ambiguous taxa which may be present as incidentals (fungal spores blown into pitchers, taxa with unknown habitat or ecological role).

Visualizing differences in predicted bacterial functions using a non-metric multidimensional scaling plot (NMDS) showed overlap of the populations but also some differences (Fig. 4B), but showed distinct functional profiles to the wastewater outgroup used (ANOSIM $p < 0.01$). Functional vectors driving separation included greater sulfur metabolism, photosynthesis and pigments, and starch and sucrose metabolism in wastewater samples, which included algae and cyanobacteria, whereas the pitcher communities were clustered with more prominent metabolism of several amino acids.

The PCoA based on eukaryotic taxa also showed overlap between the two populations (Fig. 5). Sapa samples were more different from each other than were Cedarburg samples, but both were distinct from the EUKBASE freshwater compilation. Across all samples,

**Table 1  Bacterial and eukaryotic diversity metric for two pitcher plant populations.** Summary of diversity metrics for bacterial (top) and eukaryotic (bottom) analysis of 10 *Sarracenia purpurea* pitcher fluid samples. OTUs were defined by a 97% similarity. Means of values are shown for each wetland population, bolded and *p*-values when there were significant differences between populations. There were no significantly differences in 18S data between wetlands.

### A. 16S Bacteria

| | Observed OTUs | Chao1 | singletons | Inv Simpson | Shannon | ACE | Goods Coverage |
|---|---|---|---|---|---|---|---|
| **Cedarburg** | | | | | | | |
| CB1p1 | 2664 | 8098.5 | 1688 | 16.041 | 6.155 | 7679.8 | 0.9768 |
| CB1p2 | 2923 | 10470.6 | 1993 | 17.389 | 6.446 | 9644.4 | 0.9589 |
| CB1p3 | 4186 | 14820.2 | 2975 | 19.513 | 6.591 | 15283.1 | 0.9534 |
| CB1p4 | 4326 | 22188.0 | 3434 | 43.579 | 8.149 | 23019.0 | 0.8585 |
| CB5p234 | 2731 | 8500.3 | 1759 | 14.524 | 5.512 | 8081.4 | 0.9782 |
| | **3366** | 12815.5 | 2369 | 22.209 | 6.571 | 12741.5 | 0.9452 |
| **Sapa** | | | | | | | |
| Sp3p2 | 1516 | 5153.3 | 1024 | 4.668 | 3.737 | 5153.6 | 0.9854 |
| Sp3p4 | 1592 | 5089.0 | 1078 | 2.005 | 2.721 | 5348.8 | 0.9871 |
| Sp3p5 | 1661 | 5083.7 | 1095 | 20.362 | 6.094 | 5333.2 | 0.9647 |
| Sp5p23 | 2824 | 7185.5 | 1692 | 24.648 | 6.807 | 7260.7 | 0.9714 |
| Sp9p234 | 1744 | 5164.5 | 1104 | 3.209 | 3.770 | 5043.8 | 0.9843 |
| | **1867** | **5535.2** | 1199 | 10.978 | 4.626 | **5628.0** | 0.9786 |
| | $p < 0.025$ | $p < 0.02$ | $p < 0.015$ | | | $p < 0.05$ | |

### B. 18S Eukaryotes

| | Observed OTUs | Chao1 | singletons | Inv Simpson | Shannon | ACE | Goods Coverage |
|---|---|---|---|---|---|---|---|
| **Cedarburg** | | | | | | | |
| CB1p1 | 697 | 2282.5 | 475 | 2.341 | 2.280 | 2292.7 | 0.9951 |
| CB1p2 | 978 | 3358.9 | 694 | 2.272 | 2.263 | 3558.4 | 0.9945 |
| CB1p3 | 1494 | 6493.5 | 1114 | 3.007 | 2.782 | 6521.6 | 0.9904 |
| CB1p4 | 573 | 1815.3 | 393 | 1.339 | 1.142 | 2040.3 | 0.9964 |
| CB5p234 | 2010 | 15594.1 | 1657 | 11.559 | 5.298 | 14498.1 | 0.9500 |
| | 1150 | 5908.9 | 867 | 4.104 | 2.753 | 5782.2 | 0.9853 |
| **Sapa** | | | | | | | |
| Sp3p2 | 1277 | 2854.5 | 711 | 8.945 | 4.556 | 2917.2 | 0.9924 |
| Sp3p4 | 1023 | 3354.2 | 673 | 3.654 | 3.167 | 3234.3 | 0.9939 |
| Sp3p5 | 1304 | 3408.4 | 779 | 4.762 | 4.102 | 3479.6 | 0.9904 |
| Sp5p23 | 1248 | 3317.3 | 734 | 9.183 | 4.399 | 3231.0 | 0.9943 |
| Sp9p234 | 861 | 3714.3 | 628 | 3.892 | 2.612 | 3700.6 | 0.9954 |
| | 1143 | 3329.7 | 705 | 6.087 | 3.767 | 3312.5 | 0.9933 |

eukaryotic composition of Cedarburg versus Sapa samples was not significantly different (ANOSIM, $p < 0.25$). When the most common family-level taxa were used as vectors to separate the two wetland populations in a NMDS plot (Fig. 5B), population differences were driven by ciliate groups Colpodidae, Grossglockneriidae and Chytrids which were more common in Sapa pitchers, and millipedes (Xystodesmidae), ciliates (Orchotophyridae)

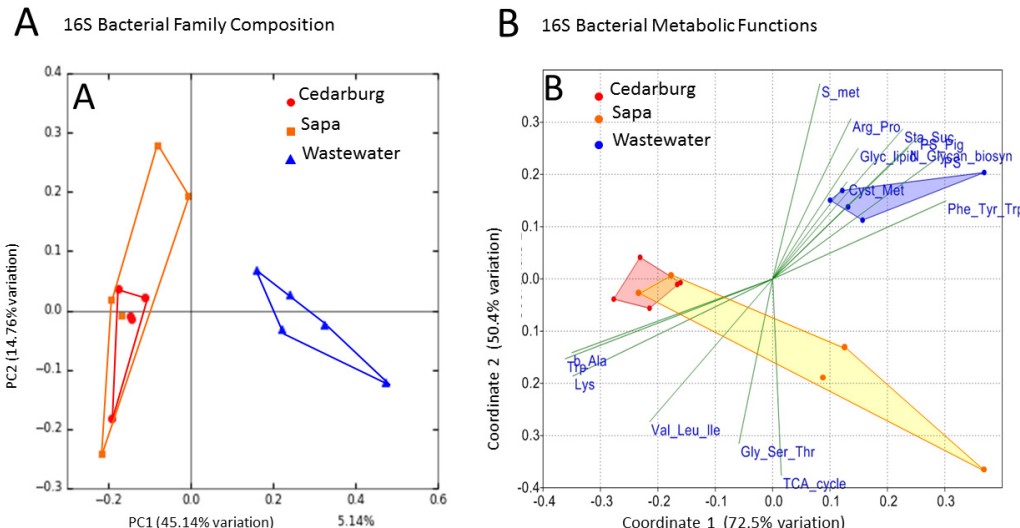

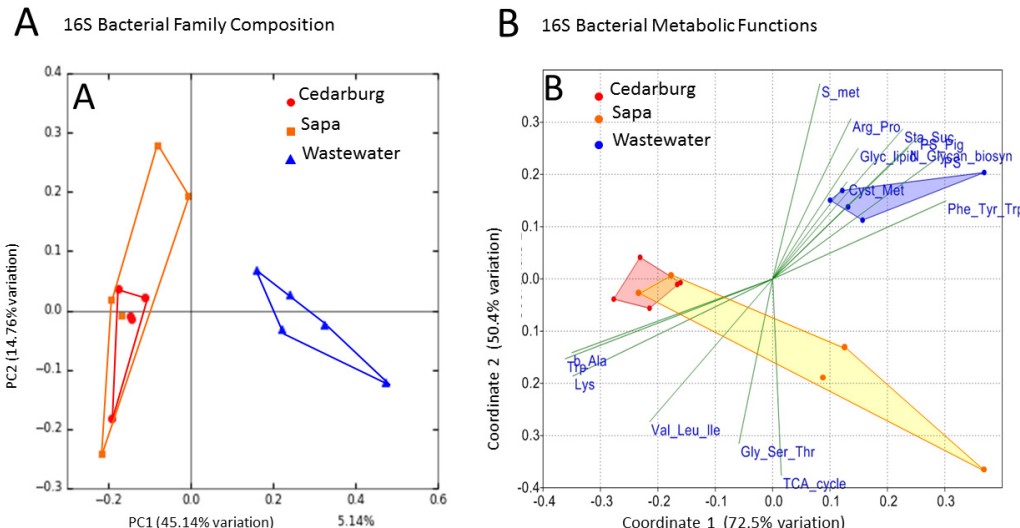

**Figure 4** **PCoA of pitcher plant bacterial composition and functions in two populations.** (A) Pitcher bacterial communities visualized with PCoA for pitcher bacterial composition, and (B) Non-metric multidimensional scaling plot of PICRUSt predicted bacterial metabolic functions, in five pitcher samples from each wetland, compared with five wastewater reference samples. Metabolic functions plot shows 15 selected function categories as vectors separating the communities. Convex hulls overlay the sample points for each group, showing overlap between the two wetland populations, which are distinct from the wastewater communities. Vector name abbreviations relate to metabolic processes associated with: b_Ala–b-Alanine, Lys–Lysine; Val_Leu-Ile–Valine-Leucine-Isoleucine; Trp–Tryptophan; Gly_Ser_Thr–Glycine-Serine-Threonine; TCA_cyle; S_met–Sulphur metabolism; Arg_Pro–Arginine-Proline; Glyc_lipid–glycerolipid; Sta_Suc–starch and sucrose; PS–photosynthesis; PS_Pig–photosynthetic pigments; Glycan_biosyn–glycan biosynthesis; Cyst_Met–cysteine-methionine; Phe_Tyr_Trp–Phenylalanine-Tyrosine-Tryptophan.

and mosquitos (Culicidae) which were more common in Cedarburg (Fig. 2, Table S2). Presence or dominance of additional ciliate, fungal or ant taxa also separated Sapa samples.

The relatedness trees show clustering of samples based on bacterial or Eukaryotic composition (Fig. 6) support the PCoA clustering and taxon composition similarities (Figs. 1 and 4A) with CB1p1 and CB1p2 showing the closest similarity (Fig. 6). However, there were no consistent clustering patterns for both bacterial and eukaryotic composition, suggesting one did not follow the other. Sp3p2 and Sp9p234 were more similar in the 16S tree and share dominance of Saccharibacteria and Comamonadaceae and absence of Coxiellaceae and Acetobacteraceae (Figs. 1 and 2), but these samples were not closely related in terms of Eukaryotic composition. Cedarburg and Sapa samples did not show distinct clustering in either tree (Fig. 6).

In comparing combined versus individual pitchers, some samples from the same plant were clearly very similar, e.g., CB1p1 and CB1p2, but others were not strikingly similar from the same plant (CB1p3 bacteria distinct from CB1p1 and CB1p2 (Fig. 6), and CB1p3 was more similar in bacterial composition to Sapa Sp5p23 and Sp3p5 (Fig. 6)). In contrast, in terms of eukaryotic composition, CB1p3 was more similar to CB1p1 and CB1p2 while CB1p4 was more similar to Sp9p234 (Fig. 6). Combined pitcher samples did not show

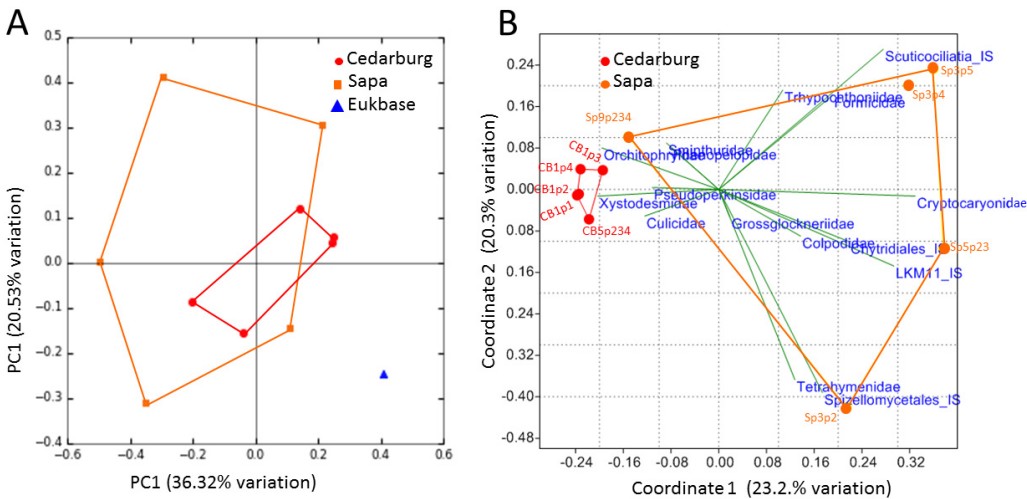

**Figure 5** **PCoA of pitcher plant eukaryotic taxa in two populations.** (A) PCoA score plot based on composition of eukaryotic taxa (18S rRNA sequences) identified in five pitcher samples from each wetland, compared with a curated freshwater Eukbase database from SILVA NR108. Convex hulls overlay the sample points for each group, showing overlap between the two wetland populations which are distinct from the Eukbase outgroup. (B) NMDS plot of community composition, using the most common 16 eukaryotic families as vectors to separate Cedarburg and Sapa samples, with correlation as the similarity measure.

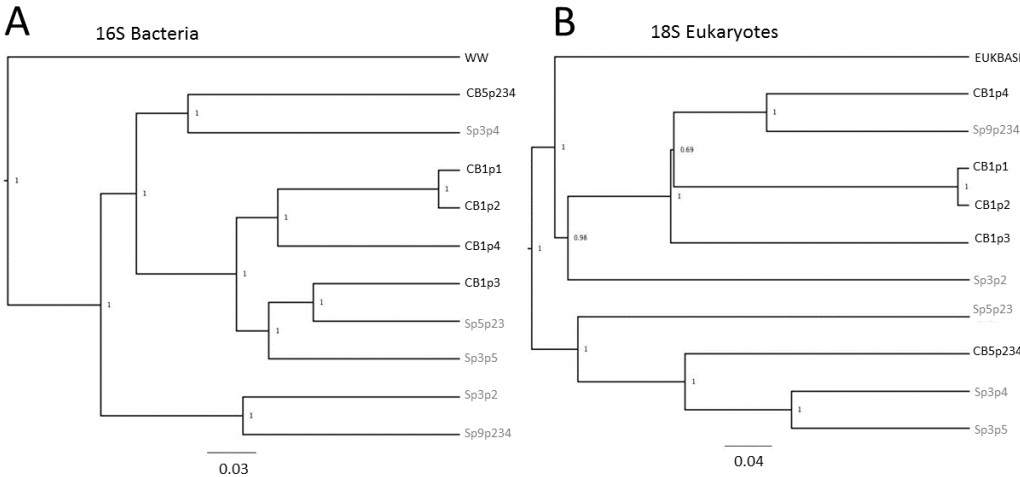

**Figure 6** **Relatedness trees for samples in two populations based on 16S and 18S rRNA sequences.** Genetic diversity trees of samples from Cedarburg Bog and Sapa Bog based on 16S bacterial (A) and 18S eukaryotic (B) composition of taxa identified in five pitcher fluid samples from each population. Bootstrap values for jackknife trees generated in QIIME were based on 100 iterations with a minimum of 75% of the smallest sample sequence number. Cedarburg (CB) sample branches are black while Sapa (Sp) sample branches are grey. Bacterial composition of samples is compared with pooled wastewater outgroup (WW), and a curated freshwater EUKBASE database from SILVA.

higher bacterial diversity than single pitcher samples—within Cedarburg samples the highest Inv Simpson, ACE and Shannon indices were for single pitcher CB1p4, and the lowest in combined pitcher CB5p234. Within Sapa bacteria, combined sample Sp9p234 had some of the lowest diversity values (Table 1).

## DISCUSSION

This study presents the first detailed characterization of composition and diversity of both bacterial and eukaryotic organisms in *S. purpurea* pitchers within and between populations, as well as analysis of bacterial functions within pitcher communities compared with another freshwater environment.

### Bacterial composition of pitcher plant communities

The dominance of Proteobacteria and Bacteriodetes in pitchers in both wetlands is similar to our preliminary genetic screening of these populations (*Young, Sielicki & Grothjan, 2018*) and to previous studies of pitcher plant bacterial communities, where $\alpha$, $\beta$ and $\gamma$-Proteobacteria were dominant and Bacteroidetes and Firmicutes were common (*Peterson et al., 2008*; *Koopman et al., 2010*; *Gray et al., 2012*; *Armitage, 2017*; *Canter et al., 2018*). Bacterial classes were the same as those typically found in wetland soils (*Zhang et al., 2017*). However, the most common phylum, Saccharibacteria AT (formerly TM7), is known in wetlands, soil and aquatic habitats (*Hugenholtz et al., 2001*), but has previously been reported only in very low abundance in pitcher plants (*Morales et al., 2006*; *Krieger & Kourtev, 2012*). The common families Sphingomonodaceae, Rhodospirillaceae, Oxalobacteraceae and many of the most abundant genera (*Pedobacter*, *Aquitalea, Sphingomonas, Rickettsiella, Azospirillum*) are common with previous reports for pitcher communities (*Gray et al., 2012*; *Northrop et al., 2017*; *Canter et al., 2018*; *Young, Sielicki & Grothjan, 2018*).

### Eukaryotic composition of pitcher plant communities

Most of the eukaryotic diversity identified in the pitchers was from taxa defined as food web inhabitants rather than prey, confirming many groups known to play roles in the *S. purpurea* food web (e.g., *Kneitel & Miller, 2002*; *Mouquet et al., 2008*; *Adlassnig, Peroutka & Lendl, 2011*). Typically important eukaryotic taxa in *S. purpurea* food web models, which were not identified from DNA sequencing, include *Wyeomyia smithii*, *Metriocnemus knabi* and *Habrotrocha rosa* (*Mouquet et al., 2008*). Previous pitcher plant surveys also have shown absence of eukaryotic groups including arachnids or algae (*Cresswell, 1991*; *Gebühr et al., 2006*). However, one might expect to encounter the key rotifer, midge and mosquito larvae predators within the pitchers sampled. Alternative chironomid (*Acricotopus*), rotifer (*Adineta vaga*) and Culicidae mosquito taxa were identified (Table S2). Identifications may relate to limited taxon representation in the SILVA database, though SILVA is still a robust option (*Balvočiūtė & Huson, 2017*). Identification of the typically sub-tropical mosquito *Aedes aegypti* also casts doubt on some genus-level identification. Manual sequence BLAST searches did not yield matches to *Wyeomyia, Metriocnemus* or *Habtrotrocha*. Lack of these taxa does raise the question, how common is it for communities in new and maturing

pitchers to lack commonly regarded 'keystone species' and how soon do pitchers typically gain these organisms?

While some expected taxa were missing, this sequencing identified an impressive and previously under-appreciated number of different ciliate, mite, fungi and flagellate taxa in pitcher communities. Mite dominance has been noted with sequencing approaches in *Sarracenia alata* and *Nepenthes* (*Bittleston et al., 2016*; *Satler, Zellmer & Carstens, 2016*) and in inquiline communities of bromeliads (*Pešić et al., 2016*). Bactivorous protozoa including flagellates and ciliates are well-documented within *S. purpurea* (*Hegner, 1926*; *TerHorst, 2011*; *Miller & Kneitel, 2005*; *Miller & TerHorst, 2012*). Dominance of *Colpoda* species, which were found in both wetlands, and *Tetrahymena* sp., which was only found in Sapa, has also been reported in pitcher plants (*Rojo-Herguedas & Olmo, 1999*; *TerHorst, 2011*) and dominant ciliates are known to influence community composition (*Paisie, Miller & Mason, 2014*; *Canter et al., 2018*). The large diversity of ciliates encountered could be related to low abundance or absence in some pitchers of mosquito larvae taxa (including *W. smithii*), which typically predate ciliates and other protozoa in pitchers; CB1p4 had the highest ciliate sequence count and lacked all mosquito taxa. The ciliate dominance and high taxonomic diversity in pitcher plants identified, especially Oligohymenophorea is more comparable to communities hosted by bromeliads (*Simão et al., 2017*).

Algae have previously been identified in Cedarburg Bog pitcher populations (*Young, Sielicki & Grothjan, 2018*) but in this study only a few algal 18S rRNA sequences were identified, and none of them typically photoautotrophic taxa, in contrast to algae identified in *Nepenthes* pitchers (*Bittleston et al., 2016*). While algal presence can vary in pitchers (*Gebühr et al., 2006*), 18S rRNA primer bias may have limited identification of algal taxa (*Bradley, Pinto & Guest, 2016*). Fungi have morphologically been identified within pitchers (*Lindquist, 1975*; *Adlassnig, Peroutka & Lendl, 2011*), and this study identified representatives of all major fungal phyla, including Chytrids (Table S2). Use of 28S rRNA sequencing targets also identified dominant fungi among OTUs from *S. alata* populations, though most of the 15 fungal OTUs named were Ascomycetes, and none were Chytrids (*Satler, Zellmer & Carstens, 2016*); a wider range of fungal groups were identified in *S. purpurea* using fungal PCR targets (*Boynton, 2012*).

## Bacterial differences within and between populations

While there were some bacterial families present in only one wetland (Fig. 2) the most abundant taxa were common to both wetlands. Bacterial composition can be very similar between pitchers, e.g., CBp1 and CBp2 on the same plant, but other pitchers on the same plant (CB1p3, CB1p4) showed different composition. Even within a plant, changes in pitcher community composition over time are likely related to pitcher age, capture of particular prey organisms and microbial colonization (*Armitage, 2017*). Pitcher age was not specifically controlled for in this study, though similar-looking pitchers were selected. Some similar and some very different bacterial composition between pitchers on just a single (CB1) plant illustrates the very variable colonization of pitchers.

Differences between the two bacterial populations may relate to habitat. The bacterial family Kiloniellaceae, only found in Sapa pitchers, has a preference for low pH (*Wiese et*

*al., 2009*), so the lower groundwater pH in Sapa (*Bott, Meyer & Young, 2008*) may provide better surrounding bog habitat as a source to colonize pitchers. In Sapa, the greater presence of the human pathogen family Neisseriaceae is intriguing, but they can also be aquatic (*Chu et al., 2018*). Higher abundance of Rhizobiales and Nitrosomonadaceae in Cedarburg may reflect more active microbial N cycling within pitchers in this habitat, which has more limiting N in wetland soils than in Sapa (*Bott, Meyer & Young, 2008*). Habitat differences are known to enrich particular bacterial taxa within pitchers (*Gray et al., 2012*; *Krieger & Kourtev, 2012*). Detailed analysis of the known habitats and functions of the bacterial taxa found in pitchers was recently published (*Young, Sielicki & Grothjan, 2018*).

## Eukaryotic differences within and between populations

The variation in Eukaryotic community composition between pitchers was greater than for bacteria, with differences in mosquito, millipede and ant taxa, and ciliates and fungi, driving NMDS separation between populations (Fig. 5). Wetland differences could relate to different surrounding plant community composition, and bog pH conditions between the two habitats (*Bott, Meyer & Young, 2008*). While both populations had mosquito, ciliate and fungal taxa, the family or genus representation often differed, suggesting different pools of species to colonize pitchers within the two wetlands. Localized air currents may also affect recruitment into pitchers; Sapa has a denser canopy and more physical obstacles between pitchers than the more open Cedarburg Bog, and morphologically, Sapa pitchers also have narrower openings which may reduce prey capture (*Bott, Meyer & Young, 2008*). Habitat and pitcher morphology may contribute to larger differences between Sapa compared with Cedarburg samples (Figs. 4 and 5) and to greater randomness of colonization by larger eukaryotes through prey capture or incidental introduction. Dominance of particular taxa may be attributable to DNA from larger multicellular individuals with many copies of the 18S rRNA gene. For example, high Diplopoda sequence counts (e.g., CB1p1) could have resulted from a single millipede within a pitcher, and presence of ant DNA (Formicidae, Table S2) would depend on rarer ant capture. Within the 5 Sapa samples, differences in presence/abundance of the freshwater ciliates Tetrahymenidae, soil fungi, Chytrids and Acari (mites) taxa, suggest that recruitment into pitchers may be rather stochastic. Many mites are parasitic on insects (*Berghoff et al., 2009*) and may be introduced with insect prey. DNA from Basidiomycete, Glomeromycota and Ascomycete taxa, present in the wetlands as plant saprophytes or symbionts, may be incidentally introduced as wind-born spores or fragments.

## Bacteria-eukaryote interactions

Eukaryotic composition, which differed between the two wetlands, including mosquito, millipede, ant, ciliate and fungal taxa, may contribute to differences in bacterial recruitment to individual pitchers. However, while samples CB1p1 and CB1p2 showed both similar bacterial and eukaryotic composition, there was no clear evidence across all samples that bacterial composition closely follows eukaryotic composition, which suggests that bacterial colonization of pitchers may not be solely related to prey capture. Colonization of more diverse bacterial communities was also not dependent on more diverse eukaryotic

representation, as Sapa samples showed higher eukaryotic diversity but lower bacterial diversity than Cedarburg samples. Early studies assumed that the largest source of bacteria within pitchers is transferred from prey (*Hepburn & St John, 1927*), and there is good experimental evidence that presence of invertebrate or protist taxa influences bacterial composition (*Peterson et al., 2008*; *Paisie, Miller & Mason, 2014*; *Canter et al., 2018*). However, contributions of bacterial taxa from prey versus wind, rain or other non-prey sources needs to be more rigorously examined.

## Bacterial taxonomic diversity

The microbial diversity calculated as Shannon diversity index can be directly compared with other studies. Shannon bacterial diversity in both populations (2.72–8.15) were generally higher than the values (2.17–2.47) for *S. purpurea* based on T-RFLP analysis (*Peterson et al., 2008*) and other inquiline communities based on DGGE (*Ponnusamy et al., 2008*), but more similar to values using Illumina sequencing reported in *S. purpurea* (*Paisie, Miller & Mason, 2014*; *Bittleston et al., 2018*) and the pitcher plant *Darlingtonia californica* (Sarraceniaceae) (*Armitage, 2017*). Shannon diversity of bacteria in these small volume pitchers was within the ranges reported for freshwater habitats (*Wang et al., 2012*; *Banerji et al., 2018*). Higher Chao1 richness in both populations (5,083–22,188) than previous reports for *S. purpurea* (∼200–500) (*Paisie, Miller & Mason, 2014*), suggests good sequence coverage, high diversity and representation of relatively rare bacterial taxa. The higher bacterial richness in Cedarburg than Sapa could be related to bog habitat conditions and plant composition (*Bott, Meyer & Young, 2008*).

## Eukaryotic taxonomic diversity

The number of Eukaryotic OTUs observed in this study are higher than seen in *S. alata* with genetic sequencing of 28S rRNA gene (*Satler, Zellmer & Carstens, 2016*) and 18S rRNA analysis in *S. purpurea* (*Bittleston et al., 2018*), and clearly much higher than possible with microscope-based analyses (*Kneitel & Miller, 2002*; *Gray, 2012*). *S. purpurea* hosts the highest diversity of eukaryotic inquilines of all pitcher plants (e.g., 10 species in *Darlingtonia california* vs 165 in *S. purpurea* (*Adlassnig, Peroutka & Lendl, 2011*), which may contribute to the higher genetic sequence diversity than in previous studies. This study contributes much higher detail of the diversity of the known types of organisms that are present and playing roles in the pitcher plant food web.

## Bacterial functions in pitcher communities

Differences in taxonomic composition between wetlands were minor relative to the comparison freshwater community, and the overlap between wetlands visualized in PCoA plots suggests the two populations supported similar communities and metabolic functions. Known bacterial functions in *S. purpurea*, mostly relate to prey degradation including extracellular hydrolytic enzyme activity of proteases, chitinases, phophatases, and cellulases as well as nutrient transformations including nitrate reduction, denitrification, and photosynthesis (*Young, Sielicki & Grothjan, 2018*). Many taxa identified in this study (Table S1), are known to have cellulolytic, chitinolytic and other hydrolytic capabilities, reviewed by *Young, Sielicki & Grothjan (2018)*. The amino-acid and carbohydrate and

glycerolipid metabolic functions identified support findings from a proteomics study which reported similar metabolic processes in pitchers artificially enriched with insect prey additions (*Northrop et al., 2017*), suggesting these are key functions for detrital breakdown in pitcher communities. Predictions from PICRUSt identified more metabolism of specific amino acids including tryptophan, alanine, and lysine, relative to the wastewater outgroup. The high frequency genus, *Duganella,* identified in all pitchers (Table S1) is known to produce anti-microbial compounds directly using tryptophan (*Choi et al., 2015*). More detailed examination of amino acid transformations in pitcher plants is warranted, particularly if plants can access amino N (*Karagatzides, Butler & Ellison, 2009*). Functional vectors also suggested that relative to the freshwater comparison communities, photosynthesis may be a minor contributor to food web C acquisition in these detrital food webs, which gain organic C from insect prey.

## SYNTHESIS AND CONCLUSIONS

This study provides the first detailed genetic analysis of eukaryotic organisms in the model food web of *S. purpurea* pitchers using mass sequencing, reporting diversity of known key taxonomic groups especially showing a remarkable and previously under-appreciated diversity of ciliates, fungi and mites in these communities. The study also compares eukaryotic and prokaryotic composition of the same pitcher samples, indicating the more stochastic nature of eukaryotic recruitment and suggesting that bacterial recruitment is not entirely linked to eukaryotic prey capture. The comparison also indicates that despite some key bacterial taxa presence differences between pitchers, prey digestion functions in pitchers are relatively preserved or converge to achieve similar food web function. Hydrolytic enzyme activity regulation in pitcher plants was similar to that in other aquatic ecosystems (*Young, Sielicki & Grothjan, 2018*), and the microbial diversity represented in these 20–30 mL pitcher communities rivals that of larger aquatic ecosystems. Key emerging questions from this study include how recruitment of bacteria versus eukaryotes into pitchers is mediated.

### Funding
Funding was provided by the University of Wisconsin-Milwaukee Chancellors award, and a Clifford Mortimer award for Limnology to Jacob Grothjan and a grant to Jacob Grothjan and Erica Young from the Great Lakes Genomic Center as a pilot test of their Mi-Seq facility. The funders had no role in study design, data collection and analysis, decision to publish, or preparation of the manuscript.

### Grant Disclosures
The following grant information was disclosed by the authors:
University of Wisconsin-Milwaukee Chancellors award.
Clifford Mortimer award.
Great Lakes Genomic Center.

## Competing Interests

The authors declare there are no competing interests.

## Author Contributions

- Jacob J. Grothjan and Erica B. Young conceived and designed the experiments, performed the experiments, analyzed the data, contributed reagents/materials/analysis tools, prepared figures and/or tables, authored or reviewed drafts of the paper, approved the final draft, collected samples.

## Field Study Permissions

The following information was supplied relating to field study approvals (i.e., approving body and any reference numbers):

Field sampling was approved by the University of Wisconsin-Milwaukee Field Station Committee.

## Data Availability

Young, Erica (2019): CB_Sp MS Sequence Archive.zip. figshare. Dataset. https://doi.org/10.6084/m9.figshare.7237649.v2.

## Supplemental Information

Supplemental information for this article can be found online at http://dx.doi.org/10.7717/peerj.6392#supplemental-information.

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
