# Peer review of "Diverse microbial communities hosted by the model carnivorous pitcher plant Sarracenia purpurea: analysis of both bacterial and eukaryotic composition across distinct host plant populations"

_PeerJ, doi:10.7717/peerj.6392_

## Round 0.1 · original submission · Major Revisions

Please provide a thorough point-by-point reply to all of the reviewers' comments. The manuscript contains a lot of infromation that are not essential to the readers and for this it needs to be shortened.

Reviewer 1 ·

Basic reporting

The manuscript is generally very well written and easy to follow. However, a few parts of the discussion were difficult to understand. In particular, lines 355-357 and 364-369. Part of this is due to the complications that surround comparing sets of taxa in a verbal way. I'd suggest that you clarify the meaning in these sections.

The literature cited is thorough and adequate. Figures and tables are informative, easy to interpret, and sufficient in number.

The manuscript is self contained while being clearly placed in the previous work.

Experimental design

The experimental design is adequate, and clearly within the scope of the journal. I have slight concerns about the research question, as this appears to be an exploratory investigation that seeks to identify the members of the community rather than question-based research. The work clearly addresses a knowledge gap, although in my mind the question of geographic variation of these communities is perhaps more interested than the question of how the communities vary as a function of the local environment.

Validity of the findings

The findings appear to be robust. The authors are careful not to over interpret their data, or to make claims beyond the findings of their work. The conclusions are clear, and speculation is left to a minimum.

Annotated reviews are not available for download in order to protect the identity of reviewers who chose to remain anonymous.

Reviewer 2 ·

Basic reporting

The Introduction and Abstract were well written and there was an excellent effort throughout the manuscript to apply relevant references. The authors clearly did a thorough literature search of the subject before writing the manuscript. The manuscript was of a professional structure, although I could not find where/if the sequences/raw data are publically available. Overall, professional English is used throughout - there are just parts throughout the text (mainly the discussion) where some sentences should be more clear. I have mentioned these points in the general comments to the authors. The results and hypotheses were relevant.

Experimental design

This research falls within the Aims and Scope of the journal. The questions are well defined and very relevant, especially in fulfilling a knowledge gap in Sarracenia research. This is a gap that was desperate to be filled and the paper will definitely contribute. The authors performed a rigorous investigation; however more information is needed in the Methods section. Specifically, Line 390-395 in the Discussion was the first time that I realized that multiple pitcher samples were taken from 1 pant. Please explain in more detail in the Methods section Line 141-142. I had assumed that one pitcher sample was taken per plant, totaling 5 plants per site. Now when I look at Table 1, I realize that 3 - 4 samples were taken from one plant in each site, and then there were 1-2 samples that were from a different plant. Please justify why the sampling was done in this manner. Other information that I would like to see in the Methods section is the depth of basepair reads the authors obtained per sample. The authors should also justify why they did not take measurements of the shape (mouth opening, etc) of the leaves that they sampled during their experiment. The size of the leaf opening could affect insect prey capture, so were the 10 leaves they selected for their study all of a similar size?

Validity of the findings

The data appears to be robust. Of course, it would be nice to have more pitcher communities sequenced, and more replicates from other plants, but this study is a good first step towards identifying the inquiline community in this model study system. Conclusions are well stated, but the Discussion is quite long and confusing and would benefit from a revision. Specific comments can be found in the general comments to the authors.

Additional comments

Overall, it is a nice study that provides information that is much needed about the Sarracenia system. It is a small study, and I have some questions about the decision to select a majority of leaves on 1 plant, and then a few others on other plants, but the results still provide an indepth description of the Sarracenia system.

For the Discussion, I became very confused with the differences in the different sub-sections. The first sub-section topic is about the microbial composition of the communities. It explains the overall similarities/differences of the two populations of Sarracenia. How is this different from the “between population” part of the second sub-section of the Discussion? Overall, I find the Discussion to be overly long (about 2 pages too long) and confusing because it appears that some information is repeated. I suggest a shortening of the Discussion, and to rework the sub-sections from 1) broad similarities/differences in composition between the two populations. Why this could be: habitat differences 2) differences within the sites (between the pitchers). Why this could be: prey differences, etc. 3) Diversity 4) Bacteria function. Try to make it clear to the readers when it is the bacterial composition being discussed, when it is the eukaryotic composition that is discussed, and when it is the combined composition that is discussed. Try not to flip back and forth between the three things within a section.

Some other minor comments throughout the manuscript are:
Line 73 : Just ciliates? What about the flagellates? I suggest rewording to say “protozoa”

Line 76-77: Maybe cite also Gebühr et al. 2006.

Line 93-93: “with higher nitrogen and phosphorus resulting in narrower pitchers….” is confusing in this sentence as I am unsure if this was a result found for one of the wetland that was used in this study or if it is an overall statement of what happened to the morphology of pitcher plants.

Line 123: “based on taxonomic and metagenomics predictions” is oddly placed in this question.

Line 235: “other protists distributed across pitcher samples” is confusing

Line 409-410: This sentence seems oddly placed the way that it is currently worded.

Line 328: “within the same S. purpurea pitchers within” is oddly worded

Line 352: “have also shown” instead of “showed”

Line 353-355: Were these species observed (morphologically) by the authors in the 10 pitcher leaves that were sampled? I think this sentence should be made stronger to re-iterate just how common it is to have these species (at least identified morphologically) across the geographic range of Sarracenia purpurea.

Line 359: within “the” 10 pitchers sampled.

Line 368: TerHorst 2001 the correct reference here?

Line 391: Suggested rewording “changes in pitcher community composition over time”

Line 394: “Some similar and some very different composition between pitchers” is oddly worded

Line 390-395: This is the first time that I realized that multiple pitcher samples were taken from 1 plant. Please explain in more detail in the Methods section Line 141-142. I had assumed that one pitcher sample was taken per plant, totaling 5 plants per site. Now when I look at Table 1, I realize that 3 - 4 samples were taken from one plant in each site, and then there were 1-2 samples that were from a different plant. Please justify why the sampling was done in this manner.

Line 394-395 again: what composition and colonization are we talking about? Just the bacteria or both the bacteria and eukaryotes? The beginning of the paragraph makes me assume we are still discussing bacterial composition, but it is not clear.

Line 395: Differences in bacterial composition? Overall, the first paragraph is confusing as I am unsure when the discussion about bacterial composition stops and the discussion about eukaryotic composition begins. Without directly looking at the NMDS figure, I first assumed that the listed eukaryote species in this sentence were affecting the bacteria composition. After reading the rest of the paragraph, I think the authors are only discussing eukaryotic composition. Please make the paragraph more clear.

Line 403: “in surrounding bog” is oddly worded

Line 408-410: “and there other known aquatic and soil habitats” is oddly worded. Plus, this sentences rather abruptly ends. I think it belongs in this paragraph but as it is presented now, it seems out of place.

Line 411-412: “the greater randomness of eukaryotic prey capture”. What happens to the similarity in composition between pitchers when the data on the eukaryotic prey is removed?

Line 527: “was found to be similar to in other…” oddly worded.

Reviewer 3 ·

Basic reporting

Line 183: cite ANOSIM statistic

Line 186: What are distance unifrac values?

Line 225: “differences in these families were not heavily influenced by a single OTU for those families” I don’t know why it would be expected that differences at such a broad level would be impacted by a single OTU.

Line 263 and Table 1: I’m not sure all of these metrics needs to be included. Do you need all of them to tell your story?

Figure 4 and 5: Wastewaster/Eukbase can be removed.

Figure 4B and 5B: The site names can be removed. Also, the labels on the metabolic function vectors should be larger.

Fig 6: Please provide bootstrap values (or other indication of branch support) on the tree.

Line 207: The naming of the samples is confusing. Are each of the labels (like CB1p1) corresponding to one of the 5 plants in each of the two sites? If so, why not just Cedarburg1, Cedarburg2, etc or CB1, CB2, etc.?

Line 282: I don’t think the outgroup is needed except for the dendrograms.

Line 284: PiCrust does not assess bacterial function.

Line 287-288: Were these functional pathways significantly different between sites? Why focus on these? Same thing in the associated figure, are all pathways included as vectors significantly different across sites? Or were all of them included?

Line 303-312: I think this section can be removed without losing anything from the story.

Experimental design

Overall, simple but OK. However, please be more clear about your sample size – this study is on 10 total pitchers? 5 from each site?

Validity of the findings

For PiCRUST – what was your NSTI score? How confidant are you with the predictions?

Line 220: Do you believe the zero abundance is real or an artifact of the methods?

Regarding the use of 18S primers to capture eukaryotic diversity, I’d like to see a bit more about the bias introduced by these primers. For example, in figure 3, is it possible that the differences in abundance between major types of eukaryotes is due to different copy numbers of 18S rRNA in their genome? I’m comfortable with a presence/absence analysis but I think given the diversity of eukaryotes focused on here more justification needs to be provided to quell concern of amplification bias across the major groups.

Similarly, has there been any study validating the use of 18S sequencing to quantify the abundance of larger eukaryotes (like mites?). If so, it would help bolster this method to be used to assess the abundance of a diverse assortment of eukaryotes.

Line 265: How can the rarefaction curves show low coverage but Good’s coverage show nearly full coverage of the diversity?

Line 360: were the keystone species missing? Or did you not pick them up with your primers? Were the pitchers visually inspected for the presence of mosquito larvae?

Line 490: comparison with the wastewater samples is irrelevant and should be removed.

Additional comments

This manuscript presents a characterization of bacterial and eukaryotic diversity within S. purpurea pitchers in two sites in Wisconsin. In addition to the comments in other sections of this review, I think this manuscript has the feel of the thesis and not a manuscript. The same story can be told if the manuscript was cut down by 25% or more. It would benefit the manuscript for the author to streamline the story and only include those data that are useful to the story to make this manuscript feel less like a “data dump”. The data are good, but the story and presentation should be streamlined.

---

## Round 0.2 · accepted · Accept

Thank you for providing a satisfactory revision.

# Reviewer 3 ·

Basic reporting

Acceptable

Experimental design

Acceptable

Validity of the findings

Acceptable

Additional comments

Nicely done on the revision.